# The COVID-19 Animal Fostering Boom: Ephemera or Chimera?

**DOI:** 10.3390/ani12101325

**Published:** 2022-05-23

**Authors:** Laura A. Reese, Jacquelyn Jacobs, Jordan Gembarski, Caden Opsommer, Bailey Walker

**Affiliations:** 1School of Planning, Design and Construction, Michigan State University, East Lansing, MI 48824, USA; 2Department of Animal Science, Michigan State University, East Lansing, MI 48824, USA; jacob175@msu.edu; 3Honors College, Michigan State University, East Lansing, MI 48824, USA; gembars2@msu.edu (J.G.); opsomme4@msu.edu (C.O.); walke818@msu.edu (B.W.)

**Keywords:** animal fostering, COVID-19, animal sheltering

## Abstract

**Simple Summary:**

There has been discussion in traditional and social media about increases in the numbers of people willing to foster animals in their homes during the pandemic. However, there is a lack of empirical data on whether that increase was a temporary response to the stress of COVID-19 or the ability to work from home, if it might have lasting effects, or indeed, whether an increase occurred at all. Using a national survey of over 600 animal shelter/rescue foster volunteers, this project answers the following research questions in the context of canine fostering. Did dog fosters increase their service during the COVID-19 pandemic? Do dog fosters intend to change their level of service as the pandemic wanes? What types of foster volunteers were most likely to increase their service during COVID-19 and plan to continue their level of service post-pandemic? Are there things organizations can do to retain fosters? The study concludes that there was an increase in fostering but that the impact is likely to be ephemeral predicated on the ability to work from home. Organizations may be able to retain foster volunteers through support, particularly emotional support, directed at the human as opposed to focusing solely on the dog.

**Abstract:**

There has been discussion in traditional and social media about increases in the numbers of people willing to foster animals in their homes during the pandemic. However, there is a lack of empirical data on whether that increase was a temporary response to the stress of COVID-19 or the ability to work from home, if it might have lasting effects, or indeed, whether an increase occurred at all. Using a national survey of over 600 animal shelter/rescue foster volunteers it appears that fostering did increase during the pandemic (x^2^ = 45.20, *p* = 0.00), particularly among volunteers working from home, those with higher education, those that were younger and male, and those that did not have their own dog. The study concludes that there was an increase in fostering but that the impact is likely to be ephemeral predicated on the ability to work from home. Organizations may be able to retain foster volunteers through support, particularly emotional support, directed at the human as opposed to focusing solely on the dog.

## 1. Introduction

There has been a good deal of discussion in traditional and social media and among animal shelter administrators about an increase in the numbers of people willing to adopt and foster animals in their homes during the pandemic [1,2]. Adoptions of animals during the lockdowns in the spring of 2020 appear to have risen sharply in the US as well as other countries such as Israel [3,4,5]. A recent study by the American Society for the Prevention of Cruelty to Animals (ASPCA) indicates that one in five households adopted a cat or a dog since the onset of COVID-19, and that the large majority of those animals were still in their homes a year later assuaging some concerns about a spate of returned animals once the pandemic started to wane [6]. The American Veterinary Medical Association also noted that pet adoption rates were increasing in 2020 [7]. Yet, transactional data reported by animal shelters on actual pet adoptions shows a more modest increase than media reports would suggest, with adoptions during 2020 the lowest in five years [8]. Surveys of shelter administrators also indicated that the number of animals adopted actually decreased between March and June of 2020 compared to the same months in 2019 [9]. Further, the relative search volume on Google for information about dog adoption decreased after July of 2020 (after peaking in April), returning to the five-year average by December [10]. This may not have been the result of lower demand, but due to limitations on shelter operations and fewer intakes and relinquishments meaning lower numbers of animals available for adoption [8,9]

The fostering of animals during this same period has also been reported to have increased [4,5,11]. The ASPCA experienced a 70% increase in animals going to foster homes; the Animal Care Centers of New York City called for 200 foster homes and received 2000 applications [12]. Nationally, foster care was reported to have risen by 43% during the lockdowns [12]. A survey by Best Friends conducted in the spring of 2020, at the height of most of the lockdowns, indicated that for 43% of respondents it was their first fostering experience, and 82% said they would continue fostering after their communities “reopened” [13]. Yet, academic research has indicated both a decrease in fostering during the pandemic in 2020 compared to 2019 as well as an increase in fostering [9,14].

There are a number of reasons to expect that animal fostering might have increased during the pandemic. First, many animal shelters reached out to their communities in an effort to increase fostering because of temporary closures and limited staff and volunteers [9]. Second, while somewhat mixed, much research has suggested that pets provide many physical and psychological benefits to humans [15]. During the stress of the pandemic, with stay-at-home orders limiting human interactions, having an animal in the home—whether through adoption or fostering—potentially increases in importance. For example, research has found that having a pet during the pandemic encouraged physical activity, enhanced feelings of normality, improved the moods of caregivers, lessened feelings of loneliness, improved mental health, and reduced stress [16,17,18,19,20]. These effects may be particularly important for the elderly and individuals with chronic health issues [21,22] and may be stronger in relation to interactions with dogs versus other species [16,23]. Pet owners indicated that their animals served as a distraction and reduced worry during COVID-19, eased the feelings of isolation, and strengthened the human–animal bond [24].

Whether fostering went up during the pandemic is still an open question due to mixed academic results, and research suggests that there may be significant variation among shelters and rescues in their fostering trends and among different types of animals, such as adult dogs versus puppies [9,14]. There is a lack of empirical data on whether any increase was a temporary response to the stress of COVID-19 or the ability to work from home, if it might have lasting effects, or indeed, whether an increase occurred at all. As Powell, et al., note, “future studies including larger sample sizes are needed to understand the true impact of the COVID-19 pandemic on shelter foster care programs” [9] (p. 9). Using a national survey of over 600 animal shelter foster volunteers, this project answers the following research questions in the context of canine fostering:Did dog fosters increase their service during the COVID-19 pandemic?Do dog fosters intend to change their level of service as the pandemic wanes?What types of foster volunteers were most likely to increase their service during COVID-19 and plan to continue their level of service post-pandemic?What types of support from shelter/rescues were most strongly related to increased fostering, i.e., are there things organizations can do to retain fosters?

This study differs from previous research in that it was conducted during the second year of the pandemic, focused on the foster volunteers themselves, and had a large number of respondents from a variety of shelters and rescues across the US. Answering the questions noted above will provide critical information to animal shelters and rescues in planning for future trends in fostering, targeting recruitment efforts to various groups, identifying the types of fosters that are most likely to be retained, and engaging in supportive practices most likely to retain fosters.

Fostering of animals in private homes as opposed to shelters is very important. Animal shelters represent a highly stressful environment for most dogs, endangering their well-being and adoptability. Stress factors include the sounds of other dogs; limited ability to exercise and restricted movement; fewer opportunities for social interactions, particularly if dogs were formally in a home; and does not allow for independent decision making which can lead to apathy [25,26,27,28]. The inability to interact with other dogs can add to the social isolation [20,25,29,30].

Foster care programs are critical to animal shelters and rescues because they: help address capacity issues; collect information about behavioral traits of the animal, particularly involving child, dog, and cat compatibility; allow sick or injured animals time and a home environment in which to heal; nurture animals too young for adoption, including socialization; and provide needed shelter breaks to allow dogs to reduce stress levels [27,31,32,33,34,35,36,37]. Living in a home setting can also lead dogs to become adoptable more quickly because they avoid the stress of the shelter [38].

Programs that allow foster parents to act as “ambassadors” with greater responsibility for getting the dogs adopted have been found to reduce return rates for adopted dogs and enlarge the geographical markets for potential adopters [39]. It is speculated that this reduction in returns may be the result of the fosters obtaining more information about the dog to pass on to the potential adopters, particularly regarding interactions with other animals and the opportunity for the adopters to see the dog in real life situations. Although such programs can increase the length of time of the adoption process, this raises little concern since the dogs are living in a home environment and gaining new skills [39]. Research has suggested that shelters and rescues that make greater use of foster programs have lower euthanasia rates [35,40,41], particularly those that make more use of volunteers for fostering [34]. Because of these benefits, it is important for organizations to be cognizant of the needs of their volunteers in terms of training, communication, support, and voice in decision making [42,43,44]. Internal shelter and rescue policies, and interactions with volunteers have been found to be critical to satisfaction and retention [45].

In summary, the purpose of this project is to empirically assess whether dog fostering increased during the pandemic and whether any increase can be expected to last beyond it. Because it is based on a survey of foster volunteers, it also seeks to identify whether there is anything animal shelters and rescues can do to retain fosters after the crisis has passed.

## 2. Materials and Methods

### 2.1. Participants

The study participants were recruited between August and December of 2021. Two organizations, the Petco and Pedigree Foundations, sent information on and a link to the survey to their lists of member shelters and rescues (2500 and 1350, respectively). The shelter and rescue representatives were asked to distribute the link to their list of dog-fostering volunteers. Because of the methods used to distribute the survey, the number of volunteers at each participating organization that received the survey is not known. To maintain anonymity, respondents were not asked to identify the organization they volunteered with. Six hundred and eleven individuals responded to the survey. A consent form began the survey, and respondents were asked to indicate whether they agreed to participate. A “yes” response allowed them to continue on to the survey questions. The study received exempt approval status from the Institutional Review Board at Michigan State University, Study 00006325.

### 2.2. Survey

The Canine Foster Care Survey was administered using Qualtrics, and all responses were recorded anonymously. This was primarily a bespoke survey due to the dearth of research exploring the needs and experiences of canine foster providers generally and on the impact of COVID-19 on foster care specifically. The survey was comprised of 29 questions with sections on: level of foster service; types of dogs fostered; stress related to fostering; satisfaction with training and support provided by the organization; reasons foster volunteers might discontinue fostering; attachment to the foster dogs; and demographics. Twenty-six questions were forced-choice, while three questions were open-ended asking respondents about procedures for the return of a foster, additional supports the fosters would have liked to have received, and advice to their organization. For this analysis questions related to extent of foster service, volunteer demographics, and the role of organization support were employed.

### 2.3. Statistical Analysis

All analyses were conducted using IBM SPSS Statistics for Windows, Version 27. Descriptive statistics were used to portray the sample and present responses to questions about fostering during COVID-19 and plans to foster after it. Crosstabulations compared general foster frequency to level of service (i.e., not in the context of COVID-19), during, and planned after the pandemic. A Person correlation was used to test for significance between general foster service, fostering during the pandemic, and plans to foster post-pandemic. Crosstabulations were also used to compare the traits of individuals who had stopped, decreased, maintained, or increased their level of service. Two-sided Pearson Chi-squares were used to assess the relationship between the level of fostering and individual traits of volunteers. Exploratory factor analysis was used as a data reduction technique for the 16 questions asking about organizational support. Finally, regression analysis was employed to explore models predicting the level of foster service during and planned after the COVID-19 pandemic.

## 3. Results

Two conceptually different organizational support factors were identified, one representing support in caring for the needs of the dog and the other organizational support for the human foster (see Table 1).

The respondents to the survey are overwhelmingly female, between the ages of 26 and 65, highly educated, live in an owned home without children under 18 but with a spouse or romantic partner, and have their own dogs. Descriptive statistics for the survey respondents are presented in Table 2 [13]. Among the responding volunteers, the largest percentage often foster dogs without special needs, followed by those with medical concerns and puppies. The lowest percentage often foster dogs with behavioral needs.

### 3.1. Canine Fostering during and Planned after COVID-19

Overall, the respondents to the survey tend to be either occasional, at 51% (seasonally, kitten season, or when not working), or frequent, at 44% (have fosters most of the time), fosters; only 5% reported being one-time fosters (see Appendix A). The level of foster service generally (i.e., not in the context of COVID-19) is significantly correlated with changes in service during the pandemic (Table 3). Specifically, the volunteers that are more frequent fosters generally were significantly more likely than more sporadic fosters to increase their fostering during COVID-19.

Table 4 shows the relationship between foster service levels during COVID-19 and planned levels after the pandemic is under control; the service during and planned after COVID-19 is significantly correlated. The foster volunteers whose level of service remained stable during the pandemic are the most likely to plan to continue at the same level after it. Those that increased their service were more likely to indicate that they planned to decrease it once the pandemic has abated. Finally, those whose service declined or stopped during COVID-19 were most likely to expect to increase it afterwards.

Table 5 presents the data on foster service levels in the context of the pandemic. Forty-nine percent of the fosters maintained the same level of service during the COVID-19 pandemic while 35% increased their service. After the pandemic, most planned to stay at the same level of fostering. In short, it appears to be the case that the fosters significantly increased their level of service during the pandemic but do not plan to sustain those levels after it.

### 3.2. Profile of Fosters with Different Service Levels

Crosstabulations were run and Pearson chi-squares calculated to explore the relationships between the traits of the foster volunteers and change in service during the pandemic (Table 6). The volunteers that significantly increased their level of fostering were younger, more highly educated, worked from home during the pandemic, and more frequently fostered dogs with medical issues. Gender was significantly related to increased fostering at the 0.06 level with the volunteers that were not female more likely to increase their service. Given the preponderance of women among the respondents this relationship should be viewed with caution.

### 3.3. Models Predicting Level of Service during and Planned after COVID-19

Two regression analyses were run to examine the correlates of increased service during COVID-19 and plans to foster after the pandemic is under control. Table 7 provides the regression results with foster service change during COVID-19 regressed on the set of independent variables measuring volunteer characteristics, the types of dogs fostered, and the two factors representing shelter or rescue support for fosters. Overall, the variables in the model account for only 14% of the variation in foster service change. The respondents that were able to work from home during the pandemic, that were more educated, and younger were significantly more likely to have increased their service. The general level of foster service is also positively and significantly correlated with the service during COVID-19; those that tend to foster more frequently generally increased their fostering during the pandemic. Finally, the fosters were significantly more likely to report increased service if their organization provided more emotional and other types of support directed at the human end of the relationship. The type of dog fostered did not remain significantly correlated to service levels in multiple regression. Based on the beta values, organizational support for volunteers is the strongest predictor of increased fostering during COVID-19.

Planned fostering post-COVID-19 is not well explained by any of the variables in the model, which only accounts for 5% of the variation in the dependent variable (Table 8). The only predictor that remains significantly correlated with future fostering plans is the volunteer’s ability to work from home.

## 4. Discussion

This study assesses the extent to which dog fostering may have increased during the COVID-19 pandemic by asking volunteers about their fostering service. It appears that there were increases in foster service during the pandemic, however, for most fosters, service levels remained unchanged. The volunteers that were more frequent fosters generally were significantly more likely than more sporadic fosters to increase their service during COVID-19. However, those that increased their service did not plan to sustain the higher level once the pandemic was under control suggesting a temporary gain for shelters but that any increase is unlikely to be sustained. Based on the significant correlation between working from home and increased levels of fostering, it is reasonable to conclude that this drop-off will occur as volunteers go back to working outside the home.

The volunteers that were most likely to have increased their level of fostering during the pandemic were more educated, younger, non-female, and without their own dogs than those that maintained the same level of service. This suggests that the pandemic may have brought in some individuals new to fostering. The volunteers that more frequently foster dogs with medical challenges were the most likely to increase their service during the pandemic, again likely because the time at home allowed for more intensive care of the dog.

While these findings suggest that fostering will drop off as volunteers go back to work (and indeed working from home was the only variable significantly related to service intentions after the pandemic), there is some evidence that shelters and rescues might be able to improve the chances that reductions in fostering will not be too drastic. The multiple regression results indicated that the most important variable in predicting whether a volunteer had increased their foster service was the organizational assistance provided to them. Support for the care of the dog is less important than that directed at the human part of the equation, specifically: providing a mentor to fosters to help them with the process; ensuring that fosters receive monetary support for taking care of the dog; providing emotional support; and maintaining high levels of communication between the foster and the organization, including information about the medical conditions that the foster may need to deal with.

There have been conflicting reports about the impact of the COVID-19 pandemic on dog adoptions generally and on fostering specifically. Media reports and some academic studies have suggested an increase in both but are likely sensitive to the point in the pandemic at which they were conducted. Empirical research suggests that increases in fostering during the pandemic may be a chimera, that it was hoped for but was in fact illusory. Shelter data do not appear to support the existence of a significant increase in fostering [9]. The current study, however, is based on responses from the fosters themselves about any increase in service. Respondents indicate that they have indeed increased their service but that the impact is likely to be ephemeral predicated as it appears to be on the ability to work from home.

## 5. Conclusions

Four questions were addressed in this study. First, it appears that foster volunteers did increase their service during the pandemic. Second, volunteers tended to expect to reduce their service after the pandemic as they went back to working outside their homes. Third, the increase in fostering during the pandemic tended to take place among volunteers who were already active prior to the pandemic. Fourth, organizations that provided greater emotional support for the human part of the fostering-dyad saw a greater increase in foster service.

These findings suggest several conclusions for animal shelters and rescues. Although the end of working from home may well also be the end of increased fostering, shelters and rescues may be able to retain some of those volunteers and/or support higher levels of fostering by volunteers who regularly foster dogs through support, particularly emotional support, directed at the human volunteer as opposed to focusing solely on the dog. Mentors may be particularly important [46]. These are more experienced fosters who provide one-on-one advice and support for newer fosters. Regular communication with volunteer fosters, particularly regarding the medical needs of the dogs, may ease anxiety about caring for animals with special health concerns. Finally, emotional support to make fosters feel less alone, help them address concerning behaviors or overcome challenging experiences, and even in dealing with compassion fatigue [47] may help ensure that volunteers remain willing to continue this critical service for animal shelters and rescues.

Future research should look more carefully at the types of support that shelters and rescues offer to their foster volunteers. More in-depth surveys, perhaps using focus groups, could explore the types of support that volunteers would find most helpful from their organizations. It would also be valuable to assess the relationship between emotional support for fosters and shelter/rescue outcomes in terms of number of adoptions, time to adoption, and save rates.

## 6. Limitations

The research has some limitations that are worth noting. First, because of the survey distribution method, it was not possible to calculate error or response rates. It was also not possible to attribute particular responses to specific shelters or rescues. Second, because the survey was conducted during the second year of the pandemic, it might be possible that some fosters had returned to work and were no longer fostering or had stepped away after the first exposure to the fostering experience. Third, the respondents are drawn from shelters and rescues across the US and thus do not necessarily represent the experiences of volunteers in other national settings.

## Figures and Tables

**Table 1 animals-12-01325-t001:** Factor analysis of questions regarding organizational support.

Factor	Factor Loading	Eigenvalue ^
Support for Care of Dog		
Received adequate training on basic handling	0.86	83.48
Received adequate training on behavior issues	0.87	8.15
Received adequate training on basic care and feeding	0.83	6.97
Received adequate training on introducing foster to resident animals	0.84	5.21
Received adequate training on health care/medical topics	0.85	4.25
Received adequate training on getting foster adopted	0.76	3.96
Received information on enrichment	0.66	3.14
Received sufficient support for training the foster	0.67	2.93
Received sufficient support on the behavioral needs of the foster	0.67	1.90
Organizational Support for Human Foster		
Provided a foster mentor for support	0.62	45.08
Received sufficient monetary support for fostering	0.58	12.66
Got adequate support from organization for adoption of dogs	0.78	11.04
Received sufficient emotional support from organization	0.78	9.43
Received information on how to contact organization in an emergency	0.64	9.08
Received enough information and support for medical conditions of fosters	0.55	6.52
There is sufficient communication between the organization and fosters	0.73	6.20

^ The factor with the largest eigenvalue has the most variance and so on, down to factors with small or negative eigenvalues that are usually omitted from the solution.

**Table 2 animals-12-01325-t002:** Descriptive characteristics of respondents (*n* = 611).

Gender	%
Female	93
Male	6
Transgender female	0
Transgender male	0.2
Gender variant/non-conforming	0.7
Not listed	0
Refused	0.7
Age	%
18–25	4
26–45	37
46–65	49
66 and over	10
Education	%
Some high school, no diploma	0.3
High school graduate	4
Some college	16
Trade/technical/vocational training	5
Associate’s degree	7
Bachelor’s degree	35
Master’s degree	23
Professional degree	4
Doctorate degree	5
Housing	%
Own single-family home	84
Own in a multi-family building	3
Rent single-family home	8
Rent in a multi-family building	4
Live with family/friends	1
Children < 18 in the home	%
0	78
1–2	19
3–4	3
More than 5	0.2
Other adults in home	%
Spouse/romantic partner	65
Other family member	9
Roommate	2
Live alone	24
Own dog	%
Yes	86
No	14
Often foster	%
Medical	23
Behavioral	15
Puppies	22
No special needs	36

**Table 3 animals-12-01325-t003:** Crosstabs, general level of foster service and service increase during COVID-19 (*n* = 569).

Level of Service during COVID-19	Level of Foster Service Generally			
	% One-Time	% Several Times per Year	% Seasonally	% Frequently
% Stopped	18.5	5.5	0.0	2.0
% Decreased	3.7	17.5	11.1	6.4
% Stayed the same	33.3	40.7	38.9	59
% Increased	44.4	36.4	50	32.5

Pearson correlation = 0.09 sig. at 0.05.

**Table 4 animals-12-01325-t004:** Crosstabs, change in level of foster service during and planned after COVID-19 (*n* = 587).

Planned Change in Level of Service Post-COVID-19	Change in Level of Service during COVID-19			
	% Stopped	% Decreased	% Stayed the Same	% Increased
% Stop	26.9	0.0	0.0	1.0
% Decrease	11.5	1.4	2.8	22.7
% Stay the same	19.2	43.5	92.7	72.4
% Increase	42.3	55.1	4.5	3.9

Pearson correlation = −0.27 sig. at 0.01.

**Table 5 animals-12-01325-t005:** Fostering during the pandemic and planned post-pandemic.

	% Stopped	% Decreased	% Stayed the Same	% Increased
Level of foster service since the onset of COVID-19	4.4	11.7	49.3	34.5
Planned level of service once pandemic has been controlled	1.5	9.8	76.6	12.0

**Table 6 animals-12-01325-t006:** Chi Square, volunteer traits and change in service during COVID-19.

Variable	X^2^	Sig (2-Sided)	N
Female	7.52	0.06	569
Age	40.56	0.00	572
Education	38.14	0.03	572
Housing	13.08	0.36	572
Children < 18	2.76	0.97	567
Other adults in home	8.38	0.50	568
Own dog	6.32	0.10	574
Work from home	27.32	0.00	464
Medical fosters	31.88	0.00	571
Behavioral fosters	17.54	0.13	556
Puppies	15.17	0.23	557
No special needs	9.86	0.63	567

**Table 7 animals-12-01325-t007:** Regression: has level of foster service changed during COVID-19?

	Unstandardized B	Standard Error	Beta	t	Sig
Worked from home	0.14	0.06	0.12	2.40	0.02
Number of children	0.01	0.08	0.01	0.18	0.86
Own home	−0.02	0.11	−0.02	−0.16	0.88
Education	0.04	0.02	0.10	1.91	0.06
Have own dogs	−0.23	0.11	−0.10	−1.98	0.05
Age	−0.22	0.06	−0.10	−1.47	0.00
Gender	−0.16	0.27	−0.08	−0.60	0.55
Level of foster service	0.13	0.04	0.17	3.02	0.00
Support for care of dog index	−0.08	0.05	−0.12	−1.77	0.14
Organizational/emotional support index	0.19	0.06	0.23	3.33	0.00
Medical fosters	−0.05	0.04	−0.06	−1.12	0.27
Behavioral fosters	−0.05	0.04	−0.07	−1.28	0.20
Puppies	0.04	0.03	0.07	1.29	0.20
No special needs	0.01	0.04	0.01	0.14	0.89
Constant	3.73	0.50		7.53	0.000
R^2^	0.14				

**Table 8 animals-12-01325-t008:** Regression: level of service expected after COVID-19 has been controlled.

	Unstandardized B	Standard Error	Beta	t	Sig
Worked from home	−0.12	0.04	−0.15	−2.95	0.00
Number of children	0.02	0.05	0.02	0.29	0.78
Own home	−0.03	0.07	−0.05	−0.34	0.74
Education	−0.01	0.02	−0.04	−0.71	0.48
Have own dogs	0.02	0.08	0.01	0.19	0.85
Age	0.02	0.04	0.03	0.43	0.67
Gender	0.02	0.18	0.02	0.12	0.90
Level of foster service	−0.02	0.03	−0.03	−0.54	0.59
Support for care of dog index	−0.01	0.04	−0.01	−0.18	0.86
Organizational/emotional support index	0.00	0.04	−0.01	−0.07	0.94
Medical fosters	0.02	0.03	0.03	0.51	0.61
Behavioral fosters	0.03	0.03	0.07	1.17	0.25
Puppies	0.03	0.02	0.08	1.41	0.16
No special needs	−0.02	0.03	−0.04	−0.68	0.50
Constant	2.93	0.34		8.70	0.00
R^2^	0.05				

## Data Availability

The data presented in this study are available on request from the corresponding author. The data are not publicly available at this time due to ongoing research projects.

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
