# Peer review of "The COVID-19 Animal Fostering Boom: Ephemera or Chimera?"

_animals, 2022, doi:10.3390/ani12101325_

Round 1

Reviewer 1 Report

Dear Authors,
the idea to conduct this research was original. In fact, you found out what was the main reason for fostering the Covid-19 dogs in the United States. Many authors before you have published papers related to dog adoption, but not to dog fostering. I hope that your results will be interesting for animal protection organizations that work to improve the welfare of ownerless dogs around the world. However, I think the paper would be even better if you attached supplementary material related to the results of your research and if you provided more explanations on statistical analysis, especially in the part related to Chi-square. I think that the main limitation of the paper is the unequal representation of the sexes. As far as I could see, there were 568 females in the sample. 

Introduction

Line 55 - [11,4,5] - Sort the numbers  in order from smallest to largest [4,5,11] 

Line 83 - As Powell, et al., note, “future studies... - Put the appropriate number of the reference after Powell et al. [9] (p. 9)

Line 106 - Instead of [25,26,27,28] it is better [25-28]

Line 107 - [25,29,20] - Sort the numbers  in order from the smallest to the largest [20,25,29]

Line 113 - [31,32,27,33,34,35,36,37] - Sort the numbers as follows [27, 31-37]

Line 124 - [40,35,41] - Sort the numbers as follows [35, 40, 41] or [35, 40-41]

Materials and Methods

Move the sentence (lines 163-164) "Two conceptually different factors were identified, one representing support in caring for the needs of the dog and the other organizational support for the human foster (Table 1)"  to the section "Results" and put it first.

Please provide more details on the statistical methods you used in the research, which relate to the chi-square and t-test. Explain which values you compared with the chi-square and which with the post hoc t-test.

At the end of the paragraph "Statistical analysis" report the p-value to which the significance level in your study was set. (The significance level was set at p < ?).

Results

The sentence (lines 172-174) "These characteristics are in line with prior
surveys of foster volunteers although the age of the respondents appears to be slightly higher [13]" from section "Results "move to section "Discussion"

Lines 180-182: "Overall, respondents to the survey tend to be either occasional 48% (seasonally, kitten season or when not working) or frequent 44% (have fosters most of the time) fosters; only 5% reported being one-time fosters." - Make a table for these results and attach it to the supplementary material. What about the other 3% of fosters? Where are they classified? The reader cannot see this information. 

I assume you applied the 4 x 4 chi-square test in Table 3. One cell contains a value of 0%. This means that the absolute value is also 0.  The chi-square statistic does not support zero as a cell value. The same remark applies to Table 4. 

Report p-value in brackets after t values for Table 5 in the text above that table:

Line 201  - ...  is significantly different (t=3.77, p=? or p<?).

Line 204 - ...  is also statistically significant (t=7.25, p=? or p<?)

Lines 213-214 "Gender was significantly related to increased fostering at the .06 level with volunteers that were not female more likely to increase their service".  My question is, is this value (p=.06)  really significant and relevant to your research?

Discussion and Conclusions

I advise you to merge the text given in the "Conclusions" section with the "Discussion" section (put it at the end of the "Discussion"). Then, in the new section “Conclusions” try to give concrete answers to your four questions, which were also the tasks of this research. Finally, try to draw a general conclusion about the significance and applicability of your study and the directions in which future research on foster care and fosters would move.

References

Describe references to the requirements of the Journal.

Reviewer 2 Report

Dear authors,

Thanks for submitting this interesting work to Animals.

Please see attached my specific comments.

Best wishes

General comment:

This is an interesting, well written work for which I recommend publication. I have some minor comments which I hope will help the authors to improve their work.

Specific comments:

ASPCA (line 40): please define the abbreviation when first used in the text.

Introduction: I would suggest concluding this paragraph with a brief statement of the study objectives (and, ideally, also with a formulated hypothesis).

Methods: It would be good to have an estimation of the number of potential participants to whom the invitation was sent, in order to calculate the response rate and have an idea of how representative the sample is of the whole population.

Line 169-171: Please check the tenses in this sentence.

Line 182-184: This sentence should be moved to the discussion as it is an explanation and not part of the study findings.

Table 1: I am not sure that all potential readers of this work will be familiar with Factor Loading Eigenvalue. Maybe it is worth adding a brief explanation in the table legend to help the reader interpreting these data.

Lines 242-245: I would avoid repetition of data in the discussion paragraph.
